

# Turbulent features of nearshore wave-current flow

Massimiliano Marino [1], Carla Faraci [2], Bjarne Jensen [3], and Rosaria Ester Musumeci [1]

[1]Department of Civil Engineering and Architecture, University of Catania, Via Santa Sofia 64, 95123, Catania, Italy
[2]Department of Engineering, University of Messina C.da di Dio, 98166, S. Agata, Messina, Italy
[3]DHI Water and Environment, Agern Allé 5, Hørsholm, Denmark

**Correspondence:** Massimiliano Marino  (massimiliano.marino@unict.it)

**Abstract.** Waves and currents influence nearly all nearshore physical processes. Their complex interaction gives birth to complex turbulence features that are far from being completely understood. In this regard, previous studies mainly focused on mean flow or inferred turbulent features from averaged velocities, seldom examining turbulent fluctuations. Moreover, the dynamics of wave-current flow have mostly been replicated in experimental channel setups, i.e. overlooking the natural occurrence of waves and long-shore currents intersecting at a near-orthogonal angle. In the present work, the hydrodynamics of near-orthogonal wave-current interaction is investigated through a physical model study. Experiments were carried out in a laboratory basin in the presence of fixed sand and gravel beds, where current only, waves only and combined flow tests were performed. Flow velocities were measured by means of Acoustic Doppler Velocimeters, through which time-, phase-averaged and turbulent velocities were obtained. Results revealed two main features of the wave-current flow. First, we observed that the superposition of waves do not necessarily induces an increase of the current bed shear stresses. Indeed, depending on bed roughness, current freestream velocity and wave orbital velocity, both enhancements or reductions of the current bed shear were observed. Moreover, application of quadrant analysis revealed a periodic evolution of the current turbulent bursts. Specifically, the number of current turbulent ejections-sweeps is reduced or increased as the wave phase progresses from antinodes to nodes and from nodes to antinodes, respectively.

## 1 Introduction

Waves and currents are usually simultaneously present in coastal waters. The turbulent activity generated by their combined flow plays a fundamental role in several physical processes such as mixing, diffusion, sediment dynamics, pollutant transport etc. (Grant and Madsen, 1986; Soulsby et al., 1993; Blondeaux, 2001). In the last decades, several studies have contributed to the present knowledge of nearshore wave-current hydrodynamics, with most of them acknowledging the strong nonlinearity of their interaction, but with widely different conclusions on how the two forcings influence their respective flow fields (Simons et al., 1988, 1993; Lodahl et al., 1998; Olabarrieta et al., 2010; Yuan and Madsen, 2014; Lim and Madsen, 2016; Zhang et al., 2022). The vast majority of those works focused almost exclusively on the mean flow, or derived turbulence properties from time-averaged velocities. In this regard, studies that investigated wave-current turbulent flow by means of turbulent velocities are very limited (Singh et al., 2016, 2018; Raushan et al., 2018; Faraci et al., 2018; Marino et al., 2020; Peruzzi et al., 2021).



Moreover, although waves and longshore currents generally cross each other with a near-orthogonal angle, most of the existing laboratory studies were conducted in wave flumes or oscillating water tunnels, i.e. with waves propagating in the same (Kemp and Simons, 1982; Simons et al., 1993; Umeyama, 2005; Yuan and Madsen, 2014) or in the opposite direction of the current (Kemp and Simons, 1983; Asano and Iwagaki, 1985; Mathisen and Madsen, 1996; Roy et al., 2018). These studies have been used over the last 50 years to validate analytical and numerical models (Grant and Madsen, 1979; Fredsøe, 1984; Styles

et al., 2017). In this regard, recent laboratory investigations on near-orthogonal wave-current interaction reported significant deviations of current velocities predicted by the abovementioned models from the experimental evidence (Fernando et al., 2011a, b; Lim and Madsen, 2016), leading to deviations of the predicted wave-altered current velocity up to 30% (Faraci et al., 2021).

    In the present work, we aim at investigating the hydrodynamics of wave-current interaction, by studying turbulence prop-

erties of a near-orthogonal combined flow field. In particular, we focused on: (i) investigating turbulence activity through turbulent velocity measurements, with a specific focus on boundary layer coherent structures; (ii) understanding the nonlinear behavior of orthogonal wave-current flow, by analyzing current bottom shear stresses and how they are altered by the superposition of surface waves. Experiments in a were carried out, in which waves and currents were generated over smooth (sand) and rough (gravel) beds. Flow velocities were measured by Acoustic Doppler Velocimeters. Measurements of turbulent velocities

were analyzed by means of quadrant analysis (Wallace, 2016), which shed light on how the superposition of waves affects the current turbulent ejection-sweep mechanism (Kim et al., 1987). Such a technique has rarely been employed in near-orthogonal wave-current flow investigations but has potential to allow a comprehensive interpretation of turbulent structures evolution.

    The paper is structured as follows: Section 2 section describes the experimental setup and plan. Section 3 section describes the methodology through which velocity time series were pre-processed and analyzed. Results of the velocity data analysis are

shown in Section 4, which are then discussed in Section 5. A conclusive section closes the work.

## 2   Experiments

A laboratory campaign was carried out at DHI Water and Environment (Hørsholm, Denmark) in a shallow water basin, in the framework of the Hydralab+ Transnational Access project WINGS (Waves plus currents INteracting at a right anGle over rough bedS), funded by the EU Commission through the Hydralab+ programme. The basin, schematized in Figure 1a, is 35.00 m x

25.00 m x 1.00 m, in the $x$, $y$ and $z$ directions respectively.



**Figure 1.** Schematization of the experimental wave basin (a); positioning of the measurement instruments (b); detail on the positioning of Acoustic Doppler Velocimeters (c)



On one side, the basin is provided with a multi-paddle piston-type 18.00 m long wavemaker. The wavemaker front is 18.00 m wide, and consists of 36 paddles, with each paddle being 1.20 m high and 0.50 m wide. The wavemaker is able to generate waves with wave height in the range $0.05 \div 0.45$ m. To reduce wave reflection, a 18.75 m barrier made up by 15 parabolic steel absorbers is positioned 12.00 m away from the wavemaker. For the same purpose, a C-shaped coarse-grained material beach is located at the side opposite to the wavemaker.

A recirculation system allows the generation of a current, which is conveyed into (out of) the basin through a 12 m inlet (outlet). An electromagnetic flowmeter with a $10^{-4}$ m$^3$s$^{-1}$ precision allowed to monitor the recirculating discharge. The still water level in the basin is measured by means of a meter stick. The bottom of the is horizontal and made of smooth concrete. In order to reproduce two different rough bottom conditions, a series of wood panels with fixed grains glued on top, were positioned on the basin floor. Specifically, sand bed (SB) and gravel bed (GB) panels, with a 50% fraction grain diameter $d_{50} = 0.0012$ m and $d_{50} = 0.025$ m respectively, were installed. The panels cover a rectangular area of 7.50 x 5.00 m, which hereinafter is called the controlled roughness area.

Water surface elevation was measured by means of 24 resistive wave gauges (WGs, Figure 1b). The wave gauges were connected to a series of analog data loggers, which allowed the adjustment of gauges resolution and sensitivity. The WGs were distributed all over the area in front of the wavemaker in order to give detailed spatial information about the wave field. Four out of 24 WGs (specifically WG11 to 14) were positioned in order to measure wave reflection coefficient by Faraci et al. (2015) method.

Flow velocities were measured by means of 5 Acoustic Doppler Velocimeters (ADVs); the model is the Vectrino+, manufactured by Nortek (Nortek, 2009). The ADVs were held together on a square chassis attached to a micrometer with a 0.0001 m precision, which allowed them to be slided vertically. The micrometer was then fixed to a bridge above the acquisition area. The distance between the ADVs in the $x$ and $y$ direction is larger than the one recommended by the manufacturer distance of 0.12 m, in order to ensure no acoustic interference between ADVs transducers and receivers. The ADVs measured velocities within a cylindrical sampling volume of 0.001 m high, with a resolution of 0.001 m/s. The accuracy is $\pm 0.5\%$ of the measured value. Sampling frequency is set equal to 100 Hz. The ADVs position are shown in Figure 1c. The performed plan of experiments is shown in Table 1.





**Table 1.** Plan of experiments. Runs 1-18 are carried out over sand bed (SB), whereas runs 19-36 over gravel bed (GB). CO = Current only, WO = Wave only, WC = Waves plus current, $h$ = water depth, $U$ = current velocity, $H$ = wave height, $T$ = wave period

| Sand bed (SB) | | | | | | Gravel bed (GB) | | | | | |
| --- | --- | --- | --- | --- | --- | --- | --- | --- | --- | --- | --- |
| Run | Type | $h$ [m] | $U$ [m/s] | $H$ [m] | $T$ [s] | Run | Type | $h$ [m] | $U$ [m/s] | $H$ [m] | $T$ [s] |
| 1 | CO | 0.40 | 0.21 | - | - | 19 | CO | 0.60 | 0.14 | - | - |
| 2 | WO | 0.40 | - | 0.18 | 2.0 | 20 | WC | 0.60 | 0.14 | 0.05 | 1.0 |
| 3 | WO | 0.40 | - | 0.12 | 2.0 | 21 | WC | 0.60 | 0.14 | 0.08 | 1.0 |
| 4 | WO | 0.40 | - | 0.08 | 2.0 | 22 | WC | 0.60 | 0.14 | 0.08 | 2.0 |
| 5 | WO | 0.40 | - | 0.08 | 1.0 | 23 | WC | 0.60 | 0.14 | 0.12 | 2.0 |
| 6 | WC | 0.40 | 0.21 | 0.18 | 2.0 | 24 | WO | 0.60 | - | 0.05 | 1.0 |
| 7 | WC | 0.40 | 0.21 | 0.12 | 2.0 | 25 | WO | 0.60 | - | 0.08 | 1.0 |
| 8 | WC | 0.40 | 0.21 | 0.08 | 2.0 | 26 | WO | 0.60 | - | 0.08 | 2.0 |
| 9 | WC | 0.40 | 0.21 | 0.08 | 1.0 | 27 | WO | 0.60 | - | 0.12 | 2.0 |
| 10 | CO | 0.60 | 0.14 | - | - | 28 | WC | 0.40 | 0.21 | 0.05 | 1.0 |
| 11 | WC | 0.60 | 0.14 | 0.08 | 2.0 | 29 | WO | 0.40 | - | 0.08 | 2.0 |
| 12 | WC | 0.60 | 0.14 | 0.12 | 2.0 | 30 | WO | 0.40 | - | 0.08 | 1.0 |
| 13 | WC | 0.60 | 0.14 | 0.18 | 2.0 | 31 | WO | 0.40 | - | 0.05 | 1.0 |
| 14 | WC | 0.60 | 0.14 | 0.08 | 1.0 | 32 | CO | 0.40 | 0.21 | - | - |
| 15 | WO | 0.60 | - | 0.08 | 2.0 | 33 | WC | 0.40 | 0.21 | 0.08 | 2.0 |
| 16 | WO | 0.60 | - | 0.08 | 1.0 | 34 | WC | 0.40 | 0.21 | 0.12 | 2.0 |
| 17 | WO | 0.60 | - | 0.12 | 2.0 | 35 | WC | 0.40 | 0.21 | 0.08 | 1.0 |
| 18 | WO | 0.60 | - | 0.18 | 2.0 | 36 | WO | 0.40 | - | 0.12 | 2.0 |

The experimental plan included current only (CO), wave only (WO) and waves plus current (WC) conditions. A total of 36 runs were carried out, Runs 1-18 over SB whereas Runs 19-36 over GB. Two different steady currents were generated by mantaining the current discharge constant ($Q$ = 1 m$^3$/s) while changing the water depth $h$ to 0.40 m or 0.60 m, corresponding to a mean current velocity of $U$ = 0.21 m/s and 0.14 m/s respectively. Froude number $Fr$ for the two current conditions is $Fr$ = 0.106 (for $U$ = 0.21 m/s) and $Fr$ = 0.058 (for $U$ = 0.14 m/s). Different regular wave conditions were considered, with wave height $H$ = 0.05 ÷ 0.18 m and wave period $T$ = 1.0 ÷ 2.0 s.

Each run consists of 16 tests, with each test having 3D velocity components measured at a different distance from bed $z$, in order to recover 16 positions along the vertical profile for each run with a specific wave - current configuration. A total of 576 tests were carried out. Measurement point distance from bed is shown in Table 2.

In order to achieve a steady current, the current recirculation system was activated 1 hour before starting the experiments. Sampling duration for CO Tests is equal to 2 minutes. Sampling duration of WO and WC Tests is 2 minutes for Tests with wave period $T$ = 1.0 s and 4 minutes for Tests with $T$ = 2.0 s, in order to collect 120 wave cycles for each test. Wavemaker is switched on 2 minutes before the start of the sampling process in order to achieve a stable wave field.





**Table 2.** ADV measurement distance from the bed $z$ for each Test.

| Test | $z$ [m] | Test | $z$ [m] |
|------|---------|------|---------|
| 1 | 0.001 | 9 | 0.025 |
| 2 | 0.002 | 10 | 0.035 |
| 3 | 0.003 | 11 | 0.050 |
| 4 | 0.005 | 12 | 0.075 |
| 5 | 0.008 | 13 | 0.120 |
| 6 | 0.011 | 14 | 0.150 |
| 7 | 0.015 | 15 | 0.200 |
| 8 | 0.020 | 16 | 0.250 |

The model validation, in terms of variability of wave height, steadiness of the current, near-orthogonality of the wave-current
flow, development of the boundary layer and other details, are thoroughly described in a previous effort by Faraci et al. (2021).

## 3   Methodology

### 3.1   Reynolds decomposition to obtain turbulent velocities

In order to obtain turbulent velocities, the measured velocity time series were decomposed into mean, phase-averaged and
turbulent components by means of Reynolds decomposition:

$$u = \bar{u} + \tilde{u} + u' \tag{1}$$

where $u$ is the measured velocity, $\bar{u}$ is the time-averaged velocity, $\tilde{u}$ is the phase-averaged velocity and $u'$ is the turbulent
(or fluctuating) velocity in the current direction. The same applies for $v$ and $w$, respectively in the wave and vertical upward
direction. The time-averaged velocities $\bar{u}$, $\bar{v}$ and $\bar{w}$, were obtained by time-averaging the instantaneous velocity time series
measured by all the ADVs. Phase-averaged velocities in the current direction, $\tilde{u}$, and wave direction, $\tilde{v}$, were computed as
follows:

$$\tilde{u} = \frac{1}{N_w} \sum_{i=1}^{N_w} (u_i - \bar{u}) \tag{2}$$

$$\tilde{v} = \frac{1}{N_w} \sum_{i=1}^{N_w} (v_i - \bar{v}) \tag{3}$$

where $N_w$ is the number of waves used for the phase-average. Waves were generated for 4 minutes for the wave period $T =$
2.0 s tests, and for 2 minutes for $T = 1.0$ s tests, in order to generate 120 waves per test, fairly larger than the minimum of 50
waves necessary for phase averaging (Sleath, 1987).





Turbulent velocities $u'$, $v'$ and $w'$, respectively in the $x$, $y$, and $w$ directions, were then obtained by subtracting the time-averaged and phase-averaged velocities from the instantaneous velocity time series.

This type of decomposition is however subject to possible contamination due to fluctuations of the wave height. In other
words, if the oscillatory flow is not perfectly regular, fluctuations of wave height will contaminate the turbulent velocity compo-nent. To account the incidence of oscillatory flow contamination, turbulent velocities were computed also using the Empirical Mode Decomposition method, or EMD (Huang et al., 1998). EMD is a promising approach to decompose signal and remove oscillatory contamination on turbulent component from a time series due to large wave height variability. The procedure of EMD decomposition used in the present work follows the one described by (Peruzzi et al., 2021), which employed EMD on
wave-current interaction experiments characterized by substantial wave height variability. Figure 2 shows turbulence intensity profiles $I_u$ (= $\sqrt{\overline{u'^2}}/\bar{u}$), which are an indication of the turbulence activity along the water column, for Run 22 (WC, $H = 0.08$ m, $T = 2.0$ s) and Run 21 (WC, $H = 0.08$ m, $T = 1.0$ s), obtained with the Reynolds decomposition (black circles) and EMD (grey crosses).

Run 22 (Figure 2a) and Run 21 (Figure 2b) have a wave height normalized standard deviation $\sigma_H/H_m$ of 0.05 and 0.16
respectively, which are respectively the smallest and the largest wave height variability of the entire dataset. Results on turbu-lence quantities calculated using Reynolds decomposition and EMD show that the two methods return very similar turbulence intensity profiles, despite the difference in wave height variability. Moreover, with the EMD method systematically gives larger values of turbulent intensity. A contamination of the turbulent velocity time series would imply Reynolds decomposition to add energy to the turbulent velocity, resulting in larger values of turbulent intensity. Instead the opposite seems to occur, with
this result being consistent for all the runs.

## 3.2  Turbulent velocity data pre-processing

Turbulent velocity data were processed in order to remove spikes in the time series. Presence of spikes is a common issue in acoustic velocimetry, and their removal (known as despiking) is considered an essential operation in velocity data processing (McLelland and Nicholas, 2000). However, a problem arises in turbulent flows as distinguishing spikes between with actual
turbulent fluctuations is crucial. Several despiking methods have been developed over the last decades (Goring and Nikora, 2002; Cea et al., 2007). The selected technique in this study is the one by Islam and Zhu (2013). This method employs a bivariate kernel-density function to generate a density map of the data, effectively isolating the turbulent data cluster from surrounding spike clusters. This technique outperforms previous methods that rely on universal noise thresholds, especially in the presence of turbulent flows, ensuring the preservation of the -5/3 slope of the turbulent velocity frequency spectrum. The
maximum percentage of removed data is 15%.

## 3.3  Space-averaged velocities

In rough flows, velocity flow fields are strictly related to the location of the point where they are measured. To investigate the spatially averaged characteristics of the flow, velocity field close to a rough boundary can be made globally homogeneous by means of space averaging. The time-averaged velocities $\bar{u}$, $\bar{v}$ and $\bar{w}$, obtained by time-averaging the instantaneous velocities





measured by all the ADVs, were furtherly averaged in order to obtain time- and space-averaged $\langle \bar{u} \rangle$, $\langle \bar{v} \rangle$ and $\langle \bar{w} \rangle$, according to the following:

$$\langle u \rangle = \frac{1}{N_{ADV}} \sum_{i=1}^{N_{ADV}} \bar{u}_i \tag{4}$$

where $N_{ADV}$ is the number of ADVs. The procedure of averaging by time and space is referred in the text as double-averaging, and allows to filter out the heterogeneous flow characteristics that depend to the specific position of the ADV.

## 3.4 Dimensional and non-dimensional parameters

Significant dimensional quantities and non-dimensional parameters were computed from double-averaged velocities to characterize the flow field. Current freestream velocity $U_c$ was computed by depth averaging the double-averaged current velocities above the expected current boundary layer upper limit (Fredsøe et al., 1999). Wave orbital velocity $U_w$ was computed by considering the phase-averaged velocity maximums at the first measurement point above the wave boundary layer thickness.

According to Fredsøe (1984), the expected wave boundary layer thickness in rough flows depends on the relative wave orbital amplitude $A_{bm}/k$, where $A_{bm}$ is the wave orbital amplitude (equal to $U_w/\omega$) and $k$ is the bottom roughness. Once the expected wave boundary layer thickness is computed, the wave orbital velocity is measured considering the lowest measurement point above the wave boundary layer thickness in the crest velocity profile. Then, the orbital velocities measured by each ADV were space-averaged in order to obtain double-averaged orbital velocity $U_w$. Once $U_c$ and $U_w$ were obtained, current and wave

Reynolds numbers were computed by the following:

$$Re_c = \frac{U_c h}{\nu}; \qquad Re_w = \frac{U_w A_{bm}}{\nu}; \tag{5}$$

where $A_{bm}$ is the wave orbital amplitude ($= U_w/\omega$, where $\omega = 2\pi/T$ is the wave frequency). A non-dimensional wave-current parameter $U_w/U_c$ was computed as an indicator of the relative importance of the waves compared to the current. Moreover, the wave-current parameter is used to distinguish two wave-current regimes: the current-dominated regime ($U_w/U_c < 1$) and

the wave-dominated regime ($U_w/U_c > 1$).

Current shear velocity $u^*$ and equivalent roughness $k_s$ were computed through a best fitting technique (Sumer, 2007). The best fit procedure is different depending whether the flow is hydraulically smooth, i.e. when the viscous sublayer thickness is larger than the bed grain size, or it is hydraulically rough, when the viscous sublayer is destroyed as the grains are larger than the supposed thickness of the viscous layer. In hydraulically smooth flow, the velocity profile in the logarithmic region follows

the law of the wall

$$\frac{u}{u^*} = \frac{1}{\kappa} ln(\frac{z u^*}{\nu}) + 5.0 \tag{6}$$



where $\kappa$ is the von Karman constant (= 0.4). In hydraulically rough flow, the near-bed velocity distribution follows the following logarithmic law:

$$\frac{u}{u^*} = \frac{1}{\kappa} \ln \frac{z}{z_o} \tag{7}$$

where $z_0 = k_s/30$, where $k_s$ is the equivalent roughness. Shear velocity was obtained from the slope of the linear fitting of $u$ and $\log z$, whereas $k_s$ was obtained through its intercept. However, an hypothesis on the position of the theoretical bottom needed to be done. The procedure follows the one suggested by Sumer (2007), by identifying between different hypotheses of theoretical bottom distance, the one that grants the larger logarithmic profile.

The computations to obtain of the shear velocity and related quantities are subject to uncertainty. Especially $k_s$ may be very different depending on the measurement chosen to be part of the logarithmic profile linear fitting. A 95% confidence interval for the slope was computed by means of a $t$-student distribution to assess the uncertainty range of the slope. A similar procedure was applied to $k_s$. From the shear velocity the Reynolds shear number was then computed, according to the relation

$$Re^* = \frac{u^* d_{50}}{\nu}; \tag{8}$$

Table 3 shows $u^*$, $k_s$ alongside their confidence interval values and $Re^*$ for all CO and WC runs.

Specifically: measured freestream current velocity $U_c$, measured orbital velocity $U_w$, wave-current regime parameter $U_w/U_c$, current Reynolds number $Re_c$ and wave Reynolds number $Re_w$. Their values are shown in Table 4. Target current velocities were chosen to have a relatively "weaker" ($Fr = 0.058$) and a "stronger" ($Fr = 0.106$) current, both in subcritical flow ($Fr <$ 1). Current Reynolds number $Re_c$ is greater than 4000 for all the experiments therefore the regime is fully turbulent, whereas 185    wave boundary layer is laminar as $Re_w < 2 \cdot 10^5$, according to Sumer et al. (2010).



**Table 3.** Shear velocity $u^*$, equivalent roughness $k_s$ and Reynolds shear number $Re^*$. Confidence intervals for $u^*$ and $k_s$ are also reported.

| Run | Bed | Type | $u^*$ [m/s] | $k_s$ [m] | $Re^*$ |
|-----|-----|------|-------------|-----------|--------|
| 1 | SB | CO | $0.0109 \pm 0.0009$ | $0.0004 \pm 0.0001$ | 13 |
| 6 | SB | WC | $0.0128 \pm 0.0021$ | $0.0029 \pm 0.0011$ | 17 |
| 7 | SB | WC | $0.0115 \pm 0.0015$ | $0.0012 \pm 0.0004$ | 15 |
| 8 | SB | WC | $0.0124 \pm 0.0013$ | $0.0022 \pm 0.0006$ | 15 |
| 9 | SB | WC | $0.0120 \pm 0.0014$ | $0.0022 \pm 0.0006$ | 14 |
| 10 | SB | CO | $0.0100 \pm 0.0012$ | $0.0141 \pm 0.0034$ | 12 |
| 11 | SB | WC | $0.0066 \pm 0.0014$ | $0.0008 \pm 0.0005$ | 8 |
| 12 | SB | WC | $0.0071 \pm 0.0011$ | $0.0011 \pm 0.0005$ | 9 |
| 13 | SB | WC | $0.0081 \pm 0.0008$ | $0.0027 \pm 0.0007$ | 10 |
| 14 | SB | WC | $0.0055 \pm 0.0014$ | $0.0003 \pm 0.0002$ | 7 |
| 19 | GB | CO | $0.0159 \pm 0.0034$ | $0.1418 \pm 0.0372$ | 398 |
| 20 | GB | WC | $0.0152 \pm 0.0015$ | $0.1299 \pm 0.0161$ | 381 |
| 21 | GB | WC | $0.0169 \pm 0.0020$ | $0.1535 \pm 0.0219$ | 421 |
| 22 | GB | WC | $0.0156 \pm 0.0018$ | $0.1316 \pm 0.0183$ | 390 |
| 23 | GB | WC | $0.0150 \pm 0.0014$ | $0.0900 \pm 0.0118$ | 375 |
| 28 | GB | WC | $0.0265 \pm 0.0054$ | $0.1006 \pm 0.0214$ | 662 |
| 32 | GB | CO | $0.0242 \pm 0.0070$ | $0.0645 \pm 0.0212$ | 606 |
| 33 | GB | WC | $0.0284 \pm 0.0024$ | $0.0877 \pm 0.0072$ | 710 |
| 34 | GB | WC | $0.0286 \pm 0.0013$ | $0.0844 \pm 0.0036$ | 716 |
| 35 | GB | WC | $0.0273 \pm 0.0031$ | $0.0698 \pm 0.0072$ | 682 |



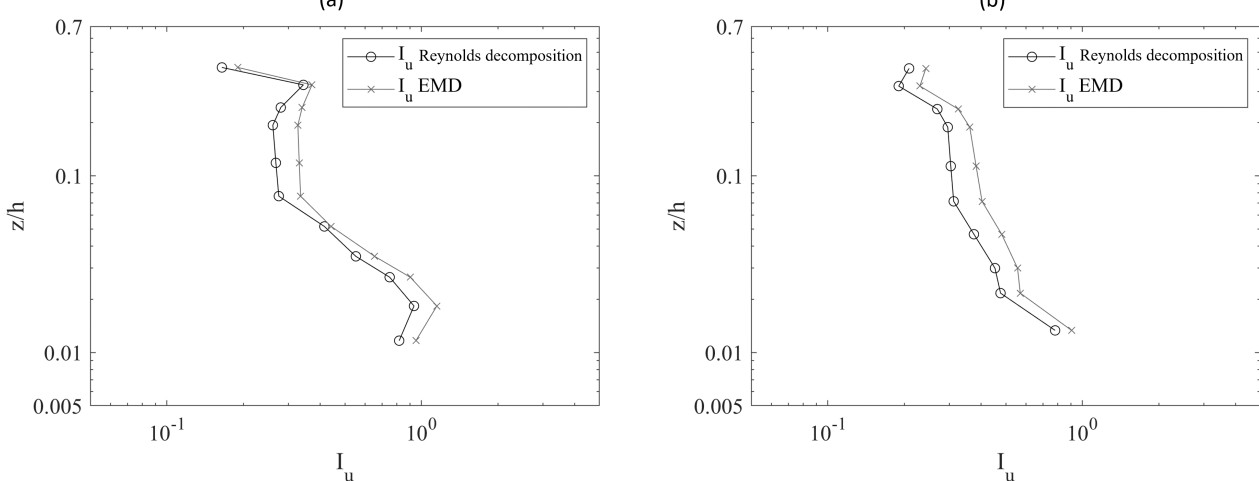

**Figure 2.** Turbulence intensity profiles $I_u$ (= $\sqrt{\overline{u'^2}}/\bar{u}$) for Run 22 (WC, $H$ = 0.08 m, $T$ = 2.0 s, a) and Run 21 (WC, $H$ = 0.08 m, $T$ = 1.0 s, b), obtained with Reynolds decomposition (black circles) and with the EMD (grey crosses)



| Run | Bed | Type | $h$ [m] | $U$ [m/s] | $H$ [m] | $T$ [s] | $U_c$ [m/s] | $U_w$ [m/s] | $U_w/U_c$ | $Fr$ | $Re_c$ | $Re_w$ |
|---|---|---|---|---|---|---|---|---|---|---|---|---|
| 1 | SB | CO | 0.40 | 0.210 | - | - | 0.226 | - | - | 0.106 | 90225 | - |
| 2 | SB | WO | 0.40 | - | 0.18 | 2.0 | - | 0.412 | - | - | - | 54031 |
| 3 | SB | WO | 0.40 | - | 0.12 | 2.0 | - | 0.325 | - | - | - | 33621 |
| 4 | SB | WO | 0.40 | - | 0.08 | 2.0 | - | 0.218 | - | - | - | 15127 |
| 5 | SB | WO | 0.40 | - | 0.08 | 1.0 | - | 0.124 | - | - | - | 2447 |
| 6 | SB | WC | 0.40 | 0.210 | 0.18 | 2.0 | 0.237 | 0.387 | 1.63 | 0.106 | 94814 | 47645 |
| 7 | SB | WC | 0.40 | 0.210 | 0.12 | 2.0 | 0.242 | 0.319 | 1.32 | 0.106 | 96923 | 32350 |
| 8 | SB | WC | 0.40 | 0.210 | 0.08 | 2.0 | 0.239 | 0.203 | 0.85 | 0.106 | 95520 | 13066 |
| 9 | SB | WC | 0.40 | 0.210 | 0.08 | 1.0 | 0.223 | 0.107 | 0.48 | 0.106 | 89255 | 1819 |
| 10 | SB | CO | 0.60 | 0.140 | - | - | 0.15 | - | - | 0.058 | 89726 | - |
| 11 | SB | WC | 0.60 | 0.140 | 0.08 | 2.0 | 0.152 | 0.146 | 0.96 | 0.058 | 91031 | 6791 |
| 12 | SB | WC | 0.60 | 0.140 | 0.12 | 2.0 | 0.157 | 0.219 | 1.39 | 0.058 | 94366 | 15236 |
| 13 | SB | WC | 0.60 | 0.140 | 0.18 | 2.0 | 0.159 | 0.313 | 1.97 | 0.058 | 95497 | 31276 |
| 14 | SB | WC | 0.60 | 0.140 | 0.08 | 1.0 | 0.137 | 0.053 | 0.39 | 0.058 | 81921 | 454 |
| 15 | SB | WO | 0.60 | - | 0.08 | 2.0 | - | 0.145 | - | - | - | 6696 |
| 16 | SB | WO | 0.60 | - | 0.08 | 1.0 | - | 0.041 | - | - | - | 270 |
| 17 | SB | WO | 0.60 | - | 0.12 | 2.0 | - | 0.212 | - | - | - | 14315 |
| 18 | SB | WO | 0.60 | - | 0.18 | 2.0 | - | 0.33 | - | - | - | 34713 |
| 19 | GB | CO | 0.60 | 0.140 | - | - | 0.142 | - | - | 0.058 | 85063 | - |
| 20 | GB | WC | 0.60 | 0.140 | 0.05 | 1.0 | 0.146 | 0.027 | 0.19 | 0.058 | 87437 | 118 |
| 21 | GB | WC | 0.60 | 0.140 | 0.08 | 1.0 | 0.153 | 0.052 | 0.34 | 0.058 | 91973 | 425 |
| 22 | GB | WC | 0.60 | 0.140 | 0.08 | 2.0 | 0.144 | 0.147 | 1.02 | 0.058 | 86552 | 6900 |
| 23 | GB | WC | 0.60 | 0.140 | 0.12 | 2.0 | 0.168 | 0.218 | 1.30 | 0.058 | 100873 | 15136 |
| 24 | GB | WO | 0.60 | - | 0.05 | 1.0 | - | 0.042 | - | - | - | 281 |
| 25 | GB | WO | 0.60 | - | 0.08 | 1.0 | - | 0.061 | - | - | - | 592 |
| 26 | GB | WO | 0.60 | - | 0.08 | 2.0 | - | 0.143 | - | - | - | 6509 |
| 27 | GB | WO | 0.60 | - | 0.12 | 2.0 | - | 0.233 | - | - | - | 17281 |
| 28 | GB | WC | 0.60 | 0.210 | 0.05 | 1.0 | 0.246 | 0.058 | 0.24 | 0.106 | 98286 | 534 |
| 29 | GB | WO | 0.40 | - | 0.08 | 2.0 | - | 0.199 | - | - | - | 12605 |
| 30 | GB | WO | 0.40 | - | 0.08 | 1.0 | - | 0.116 | - | - | - | 2142 |
| 31 | GB | WO | 0.40 | - | 0.05 | 1.0 | - | 0.061 | - | - | - | 592 |
| 32 | GB | CO | 0.40 | 0.210 | - | - | 0.245 | - | - | 0.106 | 97957 | - |
| 33 | GB | WC | 0.40 | 0.210 | 0.08 | 2.0 | 0.281 | 0.186 | 0.66 | 0.106 | 112209 | 11002 |
| 34 | GB | WC | 0.40 | 0.210 | 0.12 | 2.0 | 0.28 | 0.293 | 1.05 | 0.106 | 111937 | 27253 |
| 35 | GB | WC | 0.40 | 0.210 | 0.08 | 1.0 | 0.262 | 0.11 | 0.42 | 0.106 | 104742 | 1915 |
| 36 | GB | WO | 0.40 | - | 0.12 | 2.0 | - | 0.259 | - | - | - | 21290 |

**Table 4.** Dimensional and non-dimensional parameters for each experiment. Runs 1-18 are carried out over sand bed (SB), whereas runs 19-36 over gravel bed (GB). CO = Current only, WO = Wave only, WC = Waves plus current, $h$ is water depth, $H$ is wave height, $T$ = wave period, $U_c$ = freestream current velocity, $U_w$ wave orbital velocity, $Fr$ current Froude number, $Re_c$ current Reynolds number, $Re_w$ wave Reynolds number.



## 4   Results

### 4.1   Bed shear analysis

Current velocity profiles in the case of current only and waves plus current are compared in Figure 3 to investigate the hydro-dynamic effects of the waves superposed to the current.

Shear velocity $u^*$ and equivalent roughness $k_s$ are reported in the figure, alongside the fitting lines used for their computation (Sumer, 2007). Subscripts $_{CO}$ and $_{WC}$ indicate Current Only and Waves plus Current conditions respectively. Figure 3a, which reports a comparison between runs 1 (CO) and 8 (WC, $H$ = 0.08 m, $T$ = 2.0 s), both of them over sand bed and with $Fr$ = 0.106, shows that the superposition of the waves determines an increase of resistance experienced by the current, revealed by the increase of shear velocity by 14% and an increase of equivalent roughness by more than an order of magnitude. Analogously,

Figure 3b shows the comparison of Run 32 (CO) and Run 33 (WC, $H$ = 0.08 m, $T$ = 2.0 s) over gravel bed with $Fr$ = 0.106. With respect to the sand bed case of Figure 3a, a similar behavior is observed over gravel bed, although in this case the CO shear velocity is more than doubled in comparison with that of the sand bottom runs, due to the presence of the rough bottom. Also in this case shear velocity and equivalent roughness increase as waves are superposed to the current, by 17% and 35% respectively.

A different behavior is observed in the case of a weaker current ($Fr$ = 0.058). Figure 3c shows a comparison between Run 10 (CO) and Run 11 (WC, $H$ = 0.08 m, $T$ = 2.0 s), both of them over sand bed. In this case, the superposition of waves determines a reduction of bottom shear, with a decrease of $u^*$ of 40% and a decrease of $k_s$ of an order of magnitude. Similarly, Figure 3d illustrates a comparison between Run 19 (CO) and Run 22 (WC, $H$ = 0.08 m, $T$ = 2.0 s), over gravel bed with again $Fr$ = 0.058. In this case a slighter decrease is observed for both $u^*_{WC}$ (6%) and $k_{s,WC}$ (57%).

In order to provide an overall view on how the wave motion affects the current bottom flow, Figure 4 shows the wave-current parameter $U_w/U_c$ versus the shear Reynolds number ratio $Re^*_{WC}/Re^*_{CO}$ for all the Waves plus Current tests. The wave-current parameter indicates the relative strength of the wave motion compared to the current flow, separating the current-dominated regime ($U_w/U_c < 1$) and the wave-dominated regime ($U_w/U_c > 1$). The shear Reynolds number ratio is an expression of the shear experienced by the current relative to the current only case. $Re^*_{WC}/Re^*_{CO} < 1$ indicates a shear stress reduction

compared to the current only case, whereas $Re^*_{WC}/Re^*_{CO} > 1$ indicates a shear stress enhancement. In the presence of a "stronger" current ($Fr$ = 0.106, full markers), the superposition of the oscillatory flow always determines an increase of bed resistance, as shown by the $Re^*_{WC}/Re^*_{CO}$ being always greater than 1, no matter the bed roughness. The superposition of the laminar wave boundary layer seems to determine a stress enhancement, proved by the increase in shear velocity and equivalent roughness. This is in accordance with most of experimental evidence in the literature (Grant and Madsen, 1979; Fernando

et al., 2011a). As the relative importance of the waves increases, i.e. as $U_w/U_c$ increases, the shear seems to be enhanced in a non-monotonous fashion, with a plateau around $U_w/U_c \approx 1$.

In the presence of a "weaker" current ($Fr$ = 0.058, empty markers), a different trend is observed, since the superposition of waves determines a decrease of flow resistance compared with the current only case, as the ratio $Re^*_{WC}/Re^*_{CO}$ is always below 1. Specifically, two behaviors are reported depending on the bed roughness. Over SB, the shear increases with a linear



**Figure 3.** Velocity profiles: (a) Run 1 (CO, SB, $Fr$ = 0.106, circles) and Run 8 (WC, SB, $Fr$ = 0.106 m/s, triangles, $H$ = 0.08 m, $T$ = 2.0 s); (b) Run 32 (CO, GB, $Fr$ = 0.106, circles) and Run 33 (WC, GB, triangles, $Fr$ = 0.106, $H$ = 0.08 m, $T$ = 2.0 s); (c) Run 10 (CO, SB, $Fr$ = 0.058, circles) and Run 12 (WC, $Fr$ = 0.058, triangles, $H$ = 0.12 m, $T$ = 2.0 s); (d) Run 19 (CO, GB, $Fr$ = 0.058, circles) and Run 23 (WC, GB, $Fr$ = 0.058, triangles, $H$ = 0.12 m, $T$ = 2.0 s).

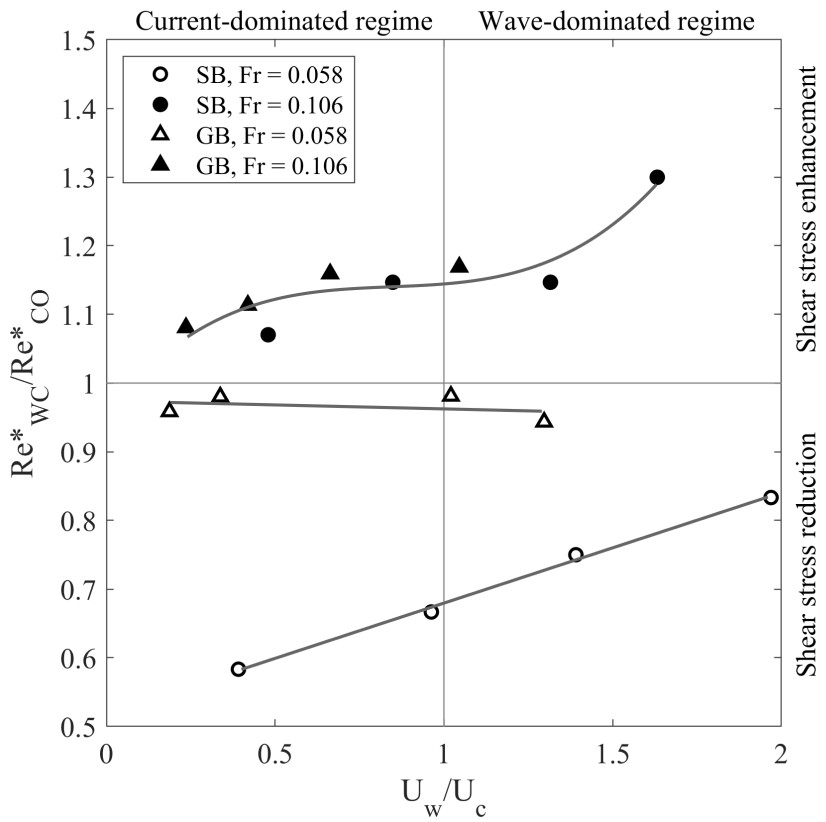

**Figure 4.** Wave-current parameter $U_w/U_c$ VS shear Reynolds number ratio $Re^*_{WC}/Re^*_{CO}$.

trend as $U_w/U_c$ increases, whereas over GB, increases the shear experienced by the current remains fairly constant as $U_w/U_c$ increases, with values of relative shear stress closer to one.

## 4.2 Turbulent flow analysis

An analysis of the turbulent velocity data was carried out and it is presented in this section. Figure 5a shows turbulence intensities $I_u$ (= $\sqrt{\overline{u'^2}}/\bar{u}$) along the current direction for Run 1 (CO), Run 7 (WC, $H$ = 0.12 m, $T$ = 2.0 s) and Run 8 (WC, $H$ = 0.08 m, $T$ = 2.0 s). All runs are over sand bed with $Fr$ = 0.106. It can be observed that the presence of the waves always enhances turbulence intensity, both close and far from the bottom. Nevertheless, even though the relative importance of the waves to the current increases, turbulence intensity profiles tends to collapse on top of each other. This is supported by the fact that the two cases have a very similar value of $Re^*_{WC}/Re^*_{CO}$.

Figure 5b shows turbulence intensities in the current direction $I_u$ for Run 32 (CO), Run 33 (WC, $H$ = 0.08 m, $T$ = 2.0 s) and Run 34 (WC, $H$ = 0.12 m, $T$ = 2.0 s) over gravel bed with the same current velocity $Fr$ = 0.106. The presence of the GB determines larger gradients of turbulence intensities, in comparison with the corresponding SB case (Figure 5a) with



**Figure 5.** Turbulence intensities $I_u$ in the current direction: (a) Run 1 (CO), Run 7 (WC, $H$ = 0.12 m, $T$ = 2.0 s, $U_w/U_c$ = 0.85) and Run 8 (WC, $H$ = 0.08 m, $T$ = 2.0 s, $U_w/U_c$ = 1.32) over sand bed with $Fr$ = 0.106; (b) Run 32 (CO), Run 33 (WC, $H$ = 0.08 m, $T$ = 2.0 s, $U_w/U_c$ = 0.66) and Run 34 (WC, $H$ = 0.12 m, $T$ = 2.0 s, $U_w/U_c$ = 1.05) over gravel bed with $Fr$ = 0.106, (c) Run 10 (CO), Run 11 (WC, $H$ = 0.08 m, $T$ = 2.0 s) and Run 13 (WC, $H$ = 0.18 m, $T$ = 2.0 s) over sand bed with $Fr$ = 0.058. (d) Run 19 (CO), Run 21 (WC, $H$ = 0.08 m, $T$ = 2.0 s) and Run 22 (WC, $H$ = 0.12 m, $T$ = 2.0 s), thus over gravel bed with $Fr$ = 0.058.



the same $U$. Notwithstanding the different wave-current regime, the two WC profiles show a very similar behavior. However, the increase of turbulence intensity in the larger $U_w/U_c$ case (Run 34) seems to extend to larger part of the water column (approximately up to $0.03 \div 0.04\ z/h$).

Figure 5c shows turbulence intensities $I_u$ in the current direction of Run 10 (CO), Run 11 (WC, $H$ = 0.08 m, $T$ = 2.0 s) and Run 13 (WC, $H$ = 0.18 m, $T$ = 2.0 s) over sand bed with $Fr$ = 0.058. As $U_w/U_c$ increases, a turbulence intensity enhancement is observed in the proximity of the bed, while a decrease in the upper part of the water column. The increase of the parameter $U_w/U_c$ seems also to affect the profile gradient in the very proximity of the bottom boundary, determining an increase of the bottom turbulence intensity gradient.

Figure 5d shows turbulence intensities $I_u$ of Run 19 (CO), Run 21 (WC, $H$ = 0.08 m, $T$ = 2.0 s) and Run 22 (WC, $H$ = 0.12 m, $T$ = 2.0 s), thus over gravel bed with $Fr$ = 0.058. The figure shows larger gradient of $I_u$ in comparison with the corresponding sand bed case with the same $U$ of Figure 5c, approximately up to $z/h$ = 0.10 m. The CO case shows a larger turbulent intensity very close to the bottom boundary, compared with all the corresponding WC cases. This could confirm the results of Figure 3d, which shows a slightly larger shear experienced by the current in the absence of waves. However, such a behavior was not observed in the SB case in Figure 5c. Moreover, slightly larger gradients of the turbulence intensity profiles are observed in the CO case.

Turbulence at a wall boundary in a steady flow is mainly generated by the succession of two cyclic events: ejections and sweeps (Corino and Brodkey, 1969). These events are the main responsible for turbulent vertical momentum transport and determine most of the generation of Reynolds shear stress (Wallace, 2016). Quadrant analysis is a well-established technique

to study the ejection-sweep cycle (Wallace et al., 1972; Lu and Willmarth, 1973; Kim et al., 1987). In quadrant analysis the turbulent events, defined as the fluctuating velocities $(u'(t), w'(t))$ at an instant $t$, where $u'$ and $w'$ are the streamwise and vertical upward direction turbulent velocities, are subdivided into four quadrants (Q1-4) depending on their signs. The second ($u' < 0$, $w' > 0$) and the fourth ($u' > 0$ and $w' < 0$) quadrants are the ones associated with the ejections and sweeps respectively. Further details on the implementation of quadrant analysis are provided in Section S1.4 of the Supplementary

document.

Figures 6a and 6b show the quadrant analysis for a Current Only case (Run 1 SB, $Fr$ = 0.106) and a Waves plus Current case (Run 8, SB, $Fr$ = 0.106, $H$ = 0.08 m, $T$ = 2.0 s) respectively, both at the same relative depth $z/h = 0.040$. The blue number inside each quadrant indicates the percentage of turbulent events that falls into the quadrant, excluding a central region defined by an hyperbolic threshold (Wallace, 2016), which excludes turbulent events not intense enough to be considered ejections or

sweeps.

Comparison between Figures 6a and 6b illustrates that the presence of waves determines an increase of intensity of turbulent activity, shown by the turbulent events being more dispersed. However, the number of turbulent events above the hyperbolic threshold is almost halved in both ejection and sweep quadrants, with the number of events inside the hyperbolic hole reaching 62%. Such a result indicates that the presence of the waves determines a decrease of the number of ejections and sweeps, but

at the same time an increase of their intensity.



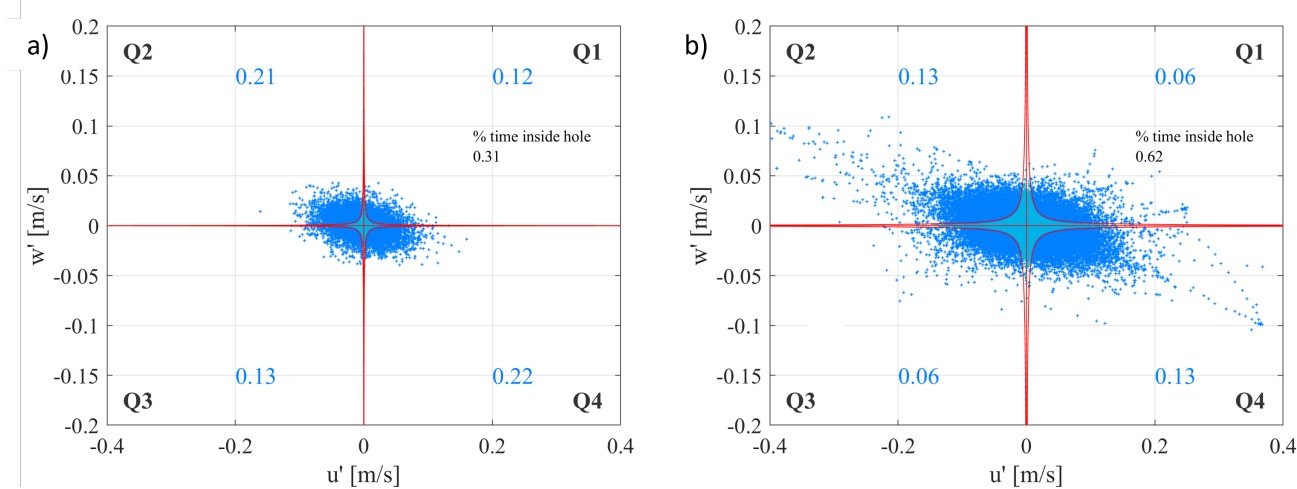

**Figure 6.** Quadrant analysis of $u'$ and $w'$ for Run 1 (Current Only, SB, $Fr$ = 0.106) (a) and Run 8 (Waves plus Current, SB, $Fr$ = 0.106, $H$ = 0.08 m, $T$ = 2.0 s) (b).

In order to observe the occurrence of turbulent bursts during a wave phase, Figure 7d illustrates the phase-averaged number of ejections and sweeps for Run 6 (WC, $H$ = 0.18 m, $T$ = 2.0 s) at $z/h$ = 0.04. Figure 7c shows the correspondent phase-averaged wave velocity.

As the wave phase progresses, a clear oscillation in the number of the turbulent bursts is observed. Turbulent bursts progres-
sively increase when the wave phase progresses from nodes to antinodes, i.e. towards crest and through phases, whereas as wave phase progresses from antinodes to nodes, a reduction of the number of ejections and sweeps is observed. The reported pattern seems to follows the nonlinearity of the wave, with the increase of number of bursts during the crest stage being shorter and more intense than the ones in the trough stage. This behavior appears to be consistent for both ejections and sweeps (black and grey line respectively). This occurrence was observed for all the runs with larger wave height cases ($H$ = 0.12 m), but not
easily recognizable for the cases with lower $H$.

## 5   Discussion

Results of our analysis highlighted that, depending on the bed roughness, freestream current velocity and wave orbital velocity, current bottom shear stress can be enhanced or reduced due to wave motion. While current shear enhancement due to wave-generated turbulence is a well-documented occurrence in literature, shear reduction is debated, although experimental evidence
supports its occurrence under specific conditions. Lodahl et al. (1998) conducted experiments in a oscillating water tunnel and measured velocity fluctuations with a Laser-Doppler anemometer in wave- and current-dominated conditions. They observe that a turbulent current can be re-laminarized by the superimposition of the oscillatory flow, with the transition being dependent on the oscillatory flow Stokes layer thickness. Lodahl and authors attributed this occurrence to the presence of the laminar wave





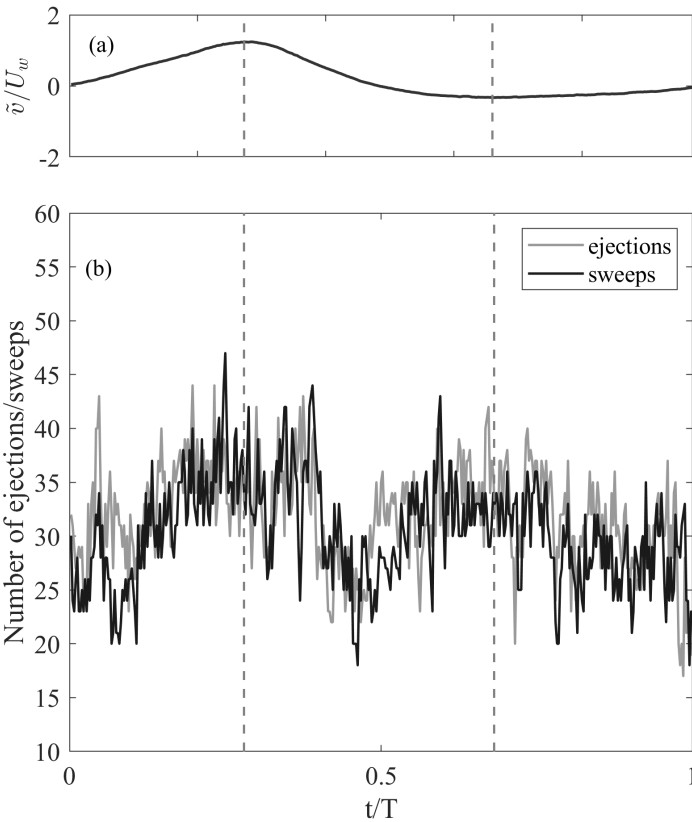

**Figure 7.** Dimensionless phase-averaged wave velocity for Run 6 ($z/h = 0.040$) (c); phase-averaged number of ejections/sweeps for Run 6, with the dashed line indicating wave crest and trough (d).

boundary layer, which is experienced by the current as a decrease of bottom hydraulic roughness. In fact, the stress patterns

reported by Lodahl and authors recall the ones shown in Figure 4. However, differently from Lodahl and authors, we clearly observed shear stress decrease also in wave-dominated conditions. In this regard, an analogous finding was reported in the near-orthogonal wave-current experiments by Musumeci et al. (2006), which attributed the phenomenon to a re-laminarization process as well, but reporting it also in wave-dominated conditions. In accordance to Musumeci and authors, shear reduction is observed only in smooth conditions also in our experiments, which resonate with analogous findings by Faraci et al. (2012),

although in a very different model setup (waves and currents interacting over a sandpit).

Another main finding of our work regards turbulence features that emerged through the application of quadrant analysis, revealing a substantial difference when the wave motion is superposed to the current with respect to the current only case. The analysis showed a reduction of the number of turbulent bursts, which was systematically accompanied by an increase of their intensity. A possible explanation involves the laminar to progressively-more-turbulent transition of the wave boundary layer.



In fact, the presence of the (temporarily) laminar wave boundary layer might induce a shear reduction which suppresses the current ejections and sweeps production. As the wave boundary layer transitions to a more turbulent state, i.e. as the wave phase progresses from nodes to antinodes, the ejection-sweep cycle resumes and the intensity of the bursts increased due to wave-generated turbulence. This interpretation is corroborated by the cyclic increase/decrease of the turbulent bursts observed in Figure 7b. This occurrence would explain both the overall decrease of number of turbulent events in the current flow, and

the increase of the turbulence intensity fluctuations.

## 6    Conclusions

In the present work, an investigation on the hydrodynamics of near-orthogonal wave-current flow turbulence was carried out through a laboratory campaign. The hydrodynamics of the wave-current flow was investigated through a comparison of the current only experiments with the ones in the presence of superposed waves. Turbulent flow properties were investigated by

studying fluctuating (turbulent) velocities. The data analysis highlighted the following main results:

- Current bed shear stress is enhanced or reduced by wave motion depending on bed roughness, current freestream velocity and wave orbital velocity. Decrease of bed shear stress is induced by the presence of the laminar wave boundary layer, which determines a decrease of shear velocity.

- The current turbulent ejection-sweep mechanism follows an oscillatory pattern determined by the superposition of the
wave motion. As the wave boundary layer develops, the number of turbulent bursts progressively increases or decreases from nodes to antinodes and from antinodes to nodes respectively.

The results of this study have implications for the modelling of sediment entrainment, suspension, and transport in nearshore environments. These findings can be implemented in analytical and numerical models to predict sediment dynamics under the influence of both waves and currents. Future experimental studies should focus on: (i) extending the range of experimented

angles and $Fr$ number, to provide an extensive analysis on the influence of the wave motion on the current velocity profile; (ii) recovering direct measurements of bottom shear stresses rather than inferring them via indirect methods, for instance by using innovative methods based on ferrofluids of bioluminescence (Foti et al., 2010; Musumeci et al., 2018; Stancanelli et al., 2020).

*Data availability.* Data will be made available on request.

*Author contributions.* MM: conceptualization, data curation, formal analysis, investigation, methodology, software, validation, visualization,
writing – original draft preparation, writing – review & editing; CF: conceptualization, funding acquisition, investigation, project Adminis-tration, supervision, writing - review & editing; BJ: Investigation, methodology, resources, supervision, writing - review & editing; REM: conceptualization, funding acquisition, investigation, methodology, project Administration, supervision, writing - review & editing



*Competing interests.* The authors declare no competing interest.

*Acknowledgements.* This work was funded by the projects: European Community's Horizon 2020 Research and Innovation Programme
through the grant to HYDRALAB-PLUS (TA WINGS - Waves plus currents INteracting at a right anGle over rough bedS), Contract no.
654110; REST-COAST - Large scale RESToration of COASTal ecosystems through rivers to sea connectivity (call: H2020-LC-GD-2020;
Proposal no. 101037097),"; National Recovery and Resilience Plan (NRRP), Mission 4 Component 2 Investment 1.3 - Call for tender No.
341 of 15/03/2022 of Italian Ministry of University and Research funded by the European Union – NextGenerationEU. Award number:
PE00000005, Concession Decree No. 1522 of 11/10/2022 adopted by the Italian Ministry of University and Research, D43C22003030002,
330   "Multi-Risk sciEnce for resilienT commUnities undeR a changiNg climate" (RETURN) - Cascade funding - Spoke VS1 "Acqua", Concession
Decree No. 2812 of 09/01/2024 adopted by the General Director of Politecnico di Milano, "Mitigation and Adaptation in Resilient Coastal
and estUarine integrated unitS" (MARCUS), "VARIO - VAlutazione del Rischio Idraulico in sistemi cOmplessi" of the University of Catania.



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
