# Peer review of "Turbulent features of nearshore wave-current flow"

_EGUsphere, 2024_

## Author Comment (AC2)

**Turbulent features of nearshore wave-current flow**
Massimiliano Marino, Carla Faraci, Bjarne Jensen and Rosaria Ester Musumeci
Response to Reviewer 2
* * *
Me and co-authors would like to thank Reviewer 2 for their comments, suggestions and insights, by which the manuscript is significantly improved. Please find enclosed the point-by-point response to reviewer's comments, with the comments indicated in bold font, the response indicated in unformatted font and manuscript quotes indicated in italic font.

**ln 38 (Experiments in a XXXXXX were carried out)     ¿ shallow water basin ?**
The missing words are indeed "shallow water basin". The typo was corrected.

**ln 58 (The bottom of the XXXXXX   is horizontal and made of soft concrete)**
The typo was corrected.

**ln 107 (respectively in the x, y and v' directions)     v' should be z**
The typo was corrected.

**ln 111 (to account the incidence)     a for is missing**
The missing "for" was added.

**ln 123-125 really difficult to understand, redo please**
Me and co-authors agree with the reviewer that the sentence could benefit from being reformulated. Lines 122-125 where substituted by the following:

*The decomposition of turbulence velocities using both Reynolds decomposition and EMD reveals that, despite variations in wave height, both methods produce similar turbulence intensity profiles. However, the EMD method consistently shows slightly higher turbulence intensity values. If the turbulent velocity time series were contaminated by Reynolds decomposition, an increase in Reynolds turbulence intensity relative to EMD would be expected. Instead, the opposite occurs: Reynolds decomposition results in lower turbulence intensity, suggesting that turbulence contamination from wave motion is not occurring. This finding is consistent across all experiments.*

**I understand that experimental numbers of table 1, have been done when considering timely order, but it makes more difficult to compare similar cases (1 vs 28). Next time re-order the lab experiments to fit within the logical comparison of the reader.**
Me and co-authors agree with the reviewer that the listing of the experiments could benefit some change. We switched run 28 (WC, GB, Fr = 0.106) with run 32 (CO, GB, Fr = 0.106), so that each nine runs block (1 CO, 4 WO, 4 WC) starts with the corresponding CO run. Thus doing, we have:
- Runs 1-9, SB, Fr = 0.106, with run 1 being CO
- Runs 10-18, SB, Fr = 0.058, with run 10 being CO
- Runs 19-27, GB, Fr = 0.058, with run 19 being CO
- Runs 28-36, GB, Fr = 0.106, with run 28 being CO

The same has not been done for WO and WC as there is not direct correspondence of wave conditions between runs (e.g. maximum wave height for sand bed is 0.18 m, whereas maximum wave height for gravel bed is 0.12 m).

**ks equivalent roughness in ln 161 is obtained from best fitting, but in line 172 is obtained through its intercept. Please clarify the discrepancy and unify the document text**

We agree that using "best fitting" and "obtained through its intercept" separately referring to the computation of $k_s$ can be misleading. To compute shear velocity $u*$ and equivalent roughness $k_s$, the best fit technique, as suggested by Sumer (2007), is used. This method involves linear fitting of the velocity profile within the logarithmic layer of a fully developed boundary layer flow. The approach varies depending on whether the flow is hydraulically smooth or rough. In hydraulically smooth flow, the velocity profile follows the law of the wall, with shear velocity obtained from the slope and $k_s$ from the intercept of the linear fit. In hydraulically rough flow, an hypothesis of the position of the theoretical bottom must be done before the linear fitting. According to Sumer (2007), the theoretical bottom should lie between 0.15÷0.30 of the physical roughness from the top of the grain. Multiple positions are tested within this range, and the one that yields the largest logarithmic layer is selected. Then, linear fitting is performed to obtain shear velocity (slope) and $k_s$ (intercept). This procedure is better explained by adding few corrections in lines 177-181:

*An hypothesis on the position of the theoretical bottom needs to be done. The procedure follows the one suggested by Sumer (2007), by identifying between different hypotheses of theoretical bottom distance, the one that grants the larger logarithmic profile. Then, shear velocity was obtained from the slope of the linear fitting of u and log (z), whereas $k_s$ was obtained through its intercept.*

**Why figure 2 is presented in vertical configuration of z/h it just make it more difficult to understand the z distance of each measurement to the floor. If it has been made to unify the figures with Figure 3, the distance to the bottom (z) could be included in the right axis of figure 2.**

A second (right side) axis indicating the corresponding dimensional depth z was added to Figure 2 in the manuscript (herein named in the supplement Figure R1).

[Figure]

Figure R1 – Turbulence intensity profiles $I_u$ for Run 22 (WC, H = 0.08 m, T = 2.0 s, a) and Run 21 (WC, H = 0.08 m, T = 1.0 s, b), obtained with Reynolds decomposition (black circles) and with the EMD (grey crosses)

**ln 200 I did not understand at the beginning the reasoning for a weaker current, include some reference to the fact that this cases are run with a larger water level.**

Me and co-authors agree with the reviewer that a clarification could be done at the point of the manuscript to better convey the reasoning behind the weaker and stronger currents. Now line 200 states the following: *In comparison with the experiments shown in figures 3a and b, the*

*experiments shown in Figure 3c and d were carried out with a larger water level (from 0.40 to 0.60 m) and keeping the discharge constant, in order to generate a weaker current (Fr = 0.058).*

**The only real comment refers to the data presented. The paper focus on the effect over turbulence of adding wave to the existing current and the reduction of turbulence bursts and increase of its intensities. It would be relevant to quantify this changes when considering the wave+current. The presented data (Figure 6) is accompanied by one number and the sentence (This occurrence was observed for all the runs with larger wave height cases, but not easily recognizable for the cases with lower H, ln 274). From my viewpoint that effect should be quantified, so that later on researchers can compare their results and add on their values to study trends under similar/different conditions.**

Some details were provided regarding the variation of turbulent features between CO and WC cases. The determinant of the covariance matrix of $u'$ and $w'$, S, is computed as an indicator of dispersion:

$$d_s = \det(S) = \sigma_{u'}^2 \sigma_{w'}^2 - \text{cov}(u', w')^2$$

where $\sigma_{u'}^2$ and $\sigma_{w'}^2$ are the variance of $u'$ and $w'$ respectively, and $\text{cov}(u', w')^2$ is the covariance of $u'$ and $w'$. While larger variances indicate more dispersion, a large covariance (whether positive or negative) indicates a stronger linear relationship between $u'$ and $w'$, which reduces the overall dispersion. Thus, the determinant decreases as covariance increases, reflecting the reduced spread in the 2D space due to the linear dependency between the variables. The determinant $d_s$ was computed for all tests and shown in Figure R2, which is added in the manuscript in a newly added Appendix section.

[Figure]

*Figure R2 – Determinant $d_s$ for all tests: SB, Fr = 0.106 tests (a); SB, Fr = 0.058 tests (b), GB, Fr = 0.106 tests (c), GB, Fr = 0.058 tests (d)*

The Figure shows that the superposition of waves determines an increase of dispersion of $(u', w')$, up to $1.5 \cdot 10^{-7}$ for SB tests (Figure R1 a and b), and up to $1.7 \cdot 10^{-6}$ for GB cases.

Except for the upper part of the water column (above approximately 0.05 m) in the case of GB, $Fr = 0.106$ (Figure R2c), superposition of waves always determines an increase of $d_s$. In line 271 of the manuscript a brief paragraph were added to provide some details about the quantification of the dispersion:

*The dispersion of u' and w' around their mean value can be quantified by computing the determinant of the covariance matrix S:*

$$d_s = det(S) = \sigma_{u'}^2 \sigma_{w'}^2 - cov(u', w')^2$$

*where $\sigma_{u'}^2$ and $\sigma_{w'}^2$ are the standard deviations of u' and w' respectively, whereas $cov(u', w')^2$ indicates the covariance of u' and w'. While larger variances indicate more dispersion, a large covariance (whether positive or negative) indicates a stronger linear relationship between u' and w', which reduces the overall dispersion. Thus, the determinant decreases as covariance increases, reflecting the reduced spread in the 2D space due to the linear dependency between the variables. The determinant $d_s$ was computed for all tests and shown in Figure A1 in Appendix A. Comparison between Figures 6a and 6b illustrates that the presence of waves determines an increase of intensity of turbulent activity, shown by the turbulent events being more dispersed, with $d_s$ increasing from $5 \cdot 10^{-8}$ to $5 \cdot 10^{-7}$.*

Regarding the quantification of the ejection-sweep oscillation during the wave phase observed in Figure 7, we agree with the reviewer that this can be quantified to allow future replication. What we observed is that, for H > 0.12 m, an oscillating behaviour is clearly observable. To quantify that, a window average on number of ejections and sweep in the wave phase (with the window being 50 Hz, in order to get the oscillation between nodes and antinodes) is carried out. Figure R3a and b shows number of ejections and sweeps per phase for Run 6. In this case, an oscillation is observed with the difference between minimum and maximum number of window averaged ejections/sweeps being 10, in comparison with the case of Figure R3c and d (Run 9, H = 0.08 m) in which the difference between minimum and maximum number of window averaged ejections/sweeps is 2.

[Figure]

*Figure R3 – Dimensionless phase-averaged wave velocity for Run 6 (a) and Run 9 (c); phase-averaged number of ejections/sweeps for Run 6 (b) and Run 9 (d), with the dashed line indicating wave crest and trough.*